# Variational Bayes Gaussian Splatting

**Toon Van de Maele**[1]
toon.vandemaele@verses.ai

**Ozan Çatal**[1]
ozan.catal@verses.ai

**Alexander Tschantz**[1,2]
alec.tschantz@verses.ai

**Christopher L. Buckley**[1,2]
christopher.buckley@verses.ai

**Tim Verbelen**[1]
tim.verbelen@verses.ai

[1] VERSES
Los Angeles, CA, USA

[2] School of Engineering and Informatics
University of Sussex
Brighton, UK

## Abstract

3D Gaussian Splatting has shown that mixture models can be used to represent high-dimensional data, such as 3D scene representations. Currently, the most prevalent method for optimizing these models is by backpropagating gradients of an image reconstruction loss through a differentiable rendering pipeline. These methods are susceptible to catastrophic forgetting in many real-world situations, where data is continually gathered through sensory observations. This paper proposes Variational Bayes Gaussian Splatting (VBGS), where we cast learning as variational inference over model parameters. Through conjugacy of the multivariate Gaussian, we find a closed-form update rule for the variational posterior, which allows us to continually apply updates from partial data, using only a single update step for each observation.

## 1 Introduction

Representing 3D scene information is a long-standing challenge for robotics and computer vision (Özyeşil et al., 2017). A recent breakthrough in this domain relies on representing the scene as a radiance field, i.e. using neural radiance fields (Mildenhall et al., 2020). Recently, Gaussian Splatting (Kerbl et al., 2023) has shown that mixture models are a competent model for this task, evidenced by a plethora of subsequent research (see Chen and Wang (2024) for a survey). This method relies on the fact that Gaussians can represent physical space as a collection of ellipsoids. The most prevalent method for optimizing the model parameters is by backpropagating gradients through a differentiable renderer, with respect to the parameters of this mixture model.

In many real-world scenarios (e.g. autonomous navigation), data is continually streamed and must be processed sequentially. Since backpropagation-based methods are susceptible to catastrophic forgetting (French, 1999), a replay buffer is typically used to retain and retrain on older data (Matsuki et al., 2024). Instead, in this paper, we cast fitting a Gaussian splat as variational inference over the parameters of a generative mixture model, for which a closed-form update rule exists (Blei et al., 2017). Continual learning is naturally enabled as these updates are a sum and can be applied iteratively. Notably, unlike gradient-based optimization, this does not require any replay buffer.

We benchmark our approach on two datasets: (i) 2D images (Tiny ImageNet (Le and Yang, 2015)), and (ii) 3D point clouds (Blender dataset (Mildenhall et al., 2020)). We evaluate these on reconstruction performance in both a setting where all data is readily available and in a continual learning setting where partial data is streamed.

Workshop on Bayesian Decision-making and Uncertainty, 38th Conference on Neural Information Processing Systems (NeurIPS 2024).

## 2 Method

The generative model considered here is a particular case of mixture models (Bishop, 2006), where each mixture component consists of two conditionally independent modalities: space ($s$) and color ($c$). For 2D images, the space component represents the pixel row and column ($s \in \mathbb{R}^2$), while for 3D data it represents the Cartesian coordinate ($s \in \mathbb{R}^3$). For both data types, we consider the color as an RGB value ($c \in \mathbb{R}^3$).

We consider both $s_k$, and $c_k$ to be distributed as multivariate Normal distributions with parameters $(\mu_{s,k}, \Sigma_{s,k})$, and $(\mu_{c,k}, \Sigma_{c,k})$ respectively. The components have a mixture weight $z$, distributed as a Categorical with parameters $\pi$. Considering these parameters as latent random variables allows us to cast learning as inference. Specifically, $(\mu_{s,k}, \Sigma_{s,k})$ are distributed according to a Normal Inverse Wishart distribution, $\mu_{c,k}$ is distributed as a Normal distribution, and $\Sigma_{c,k}$ as a delta distribution. The prior over the component weight is a Dirichlet distribution. Appendix C.1 contains a table with the used hyperparameters. The joint distribution is factorized as:

$$p(s, c, z, \mu_s, \Sigma_s, \mu_c, \Sigma_c, \pi) = \left( \prod_{n=1}^{N} p(s_n|z_n, \mu_s, \Sigma_s) p(c_n|z_n, \mu_c, \Sigma_c) p(z_n|\pi) \right) \tag{1}$$

$$\left( \prod_{k=1}^{K} p(\mu_{k,s}, \Sigma_{k,s}) p(\mu_{k,c}, \Sigma_{k,c}) \right) p(\pi). \tag{2}$$

During parameter inference, we aim to find the posterior over model parameters which maximizes model evidence concerning the observed data $(s, c)$. However, as computing this posterior is intractable, we resort to variational inference (Jordan et al., 1998). We introduce a variational posterior (denoted by q) with the following mean-field approximation

$$q(z, \mu_s, \Sigma_s, \mu_c, \Sigma_c, \pi) = \left( \prod_{n=1}^{N} q(z_n) \right) \left( \prod_{k=1}^{K} q(\mu_{k,c}, \Sigma_{k,c}) \right) \left( \prod_{k=1}^{K} q(\mu_{k,s}, \Sigma_{k,s}) \right) q(\pi), \tag{3}$$

We consider each of these distributions from the same family as the generative model described above. We optimize the evidence lower bound (ELBO) with respect to the variational parameters (Jordan et al., 1998)

$$\text{ELBO} = \mathbb{D}_{\text{KL}}[q(z, \mu_s, \Sigma_s, \mu_c, \Sigma_c, \pi) \,||\, p(z, \mu_s, \Sigma_s, \mu_c, \Sigma_c, \pi|s, c)], \tag{4}$$

using coordinate ascent variational inference (Beal, 2003; Bishop, 2006; Blei et al., 2017). This method leverages the mean field factorization and alternates updating the model assignments $q(z)$, using the variational posterior over parameters $q(\mu_s, \Sigma_s, \mu_c, \Sigma_c, \pi)$ from the previous step, and vice versa. As each of the conjugate priors are conjugate, and part of the exponential family, computing a model update reduces to summing the sufficient statistics of the data together with their prior's natural parameters (see Appendix B for the update rules).

This optimization scheme naturally lends itself to continual learning, as the parameter updates for each component are computed as the sum of the prior's natural parameters and the sufficient statistics of the data associated with this component, through assignments $q(z)$. These updates are inherently order invariant and therefore allow for iterative updates. Note that components without assignments revert to their prior. Crucially, in this setting, the assignments $q(z)$ should always be computed with respect to the parameters of the *initial* approximate posterior over the parameters $q(\mu_s, \Sigma_s, \mu_c, \Sigma_c, \pi)$. This ensures that components without prior assignments can still be used to model the data.

Images are generated using the mixture model by computing the expected value of the color, conditioned on a spatial coordinate ($\mathbb{E}_{p(c|s)}[c]$). For 3D rendering, we use the renderer from Kerbl et al. (2023), where the spatial component is first projected onto the image plane using the camera parameters, and the estimated depth is used to deal with occlusion. For more details, see Appendix D.

## 3 Results

We benchmark our approach against backpropagating gradients through a differentiable renderer. In particular, we compare the following models on both the Tiny ImageNet (Le and Yang, 2015) and Blender 3D (Mildenhall et al., 2020) datasets:

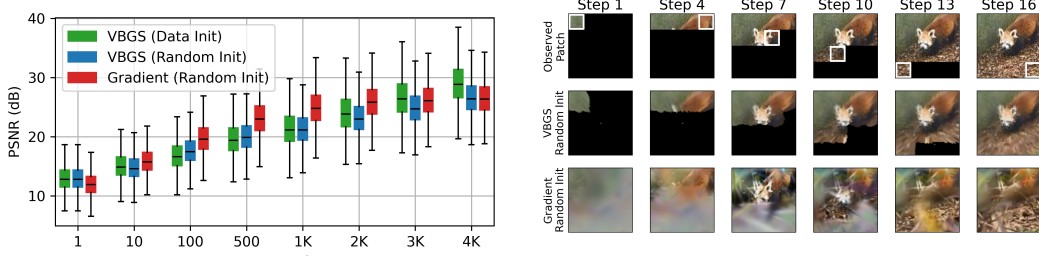

Figure 1: **Mixture model performance on image data.** (left) Shows the reconstruction performance, in PSNR (dB), for various numbers of components. (right) Shows reconstruction performance, in PSNR (dB), at various stages in a continual learning setting for a random initialized model (both VBGS and Gradient-based) with 1K components.

**VBGS (Ours)**: We consider a variant of VBGS where the means of the initial posteriors $(q(\mu_s, \Sigma_s, \mu_c, \Sigma_c))$ are initialized on sampled points from the normalized data set (Data Init), as well as randomly initialized ($m_{k,s} \sim \mathcal{U}[-1, 1]$, $m_{k,c} \sim \delta(0)$) (Random Init). Note that in the case of random initialization, the data is first normalized using the estimated statistics, refer to Appendix E for more details.

**Gradient**: In 3DGS (Kerbl et al., 2023) the parameters are directly optimized using stochastic gradient descent on a weighted image reconstruction loss $((1 - \lambda) \cdot \mathrm{MSE} + \lambda \cdot \mathrm{SSIM})$. In the case of image data, $\lambda$ is set to 0, in the case of 3D data $\lambda$ it is set to 0.2. We use spherical harmonics with no degrees of freedom, i.e., the specular reflections are not modeled. In order to be able to compare performance w.r.t. model size, we also don't do densification or shrinking (Kerbl et al., 2023) (for these results see Appendix F.3). When optimizing for images, we use a fixed camera pose at identity and keep the z-coordinate of the Gaussians fixed at a value of 1. Similar to the VBGS approach, we also consider a variant where the means of the Gaussian components are randomly initialized (Random Init), and on sampled points from the dataset (Data Init).

We first evaluate performance on the Tiny ImageNet test set (10k images). We measure the reconstruction accuracy in terms of PSNR (dB) for varying numbers of components (Figure 1 (left)). We observe that our approach performs on par in terms of reconstruction error. Next, we evaluate the model in a continual learning setting: here, patches of data are sequentially observed and processed (Figure 1 (right)). While VBGS can represent all the previously observed patches with an equal level of quality, the gradient-based approach emphasizes the last observed patch. For a quantitative analysis of the performance of continual learning over the entire validation set, see Appendix F.2.

As an additional experiment, we evaluated the wall-clock time it takes for the Gradient approach to reach the performance that VBGS achieves after a single update step. This was computed over all images of the Tiny ImageNet validation set. We observe that VBGS is significantly (t-test, $p = 0$) faster in wall clock time ($0.03 \pm 0.03$ seconds) compared to Gradient ($0.05 \pm 0.02$ seconds).

Next, we evaluate VBGS for 3D structures using the Blender dataset from Mildenhall et al. (2020). We fit model parameters using the 200 frames from the test set because this also contains depth information. VBGS is trained on the 3D point cloud, which is acquired by transforming the RGBD frame to a shared reference frame. In contrast, the gradient-based approach is optimized using multi-view image reconstruction. We evaluate reconstruction performance on 100 frames from the validation set. The results, measured as PSNR, for a model with a capacity of 100K components are shown in Table 1. We observe that both approaches perform better when the components are initialized using the data instead of randomly. We find that our approach performs on par with the gradient-based approach for Data Init. In random initialization, our approach outperforms the gradient-based approach except for the "ship" object. For a more in-depth analysis where we vary the number of components, see Appendix F.1.

Novel view predictions for the 8 blender objects are shown in Figure 2 (left). These renders are generated using a VBGS with 100K components, observed from a camera pose selected from the validation set. Note that these are rendered on a white background, and only the 3D object is modeled by the VBGS. Figure 2 (right) shows the reconstruction performance as a function of the number of available components. It can be observed that for lower component regimes, VBGS renders patches

Table 1: **Reconstruction performance for the 3D dataset**. Measured as PSNR (dB). Values ($\mu \pm \sigma$) are computed over 100 validation frames for each of the 8 blender objects. All models in this table have 100K components. The best performance for each column is marked in bold.

| | chair | drums | ficus | hotdog | lego | materials | mic | ship |
|---|---|---|---|---|---|---|---|---|
| VBGS (Data Init) | 22.82 ±0.94 | **19.50** ±0.47 | **22.06** ±0.79 | **23.62** ±1.23 | **22.53** ±0.89 | **20.55** ±1.64 | 24.78 ±0.60 | 21.23 ±0.64 |
| VBGS (Random Init) | 21.35 ±0.69 | 18.71 ±0.43 | 21.49 ±0.75 | 22.24 ±0.97 | 20.59 ±0.86 | 20.47 ±1.36 | 24.42 ±0.58 | 20.80 ±0.90 |
| Gradient (Data Init) | **22.98** ±1.23 | 19.05 ±0.57 | 21.08 ±0.92 | 21.47 ±1.67 | 19.97 ±2.24 | 20.53 ±1.64 | **25.25** ±0.90 | **23.55** ±1.20 |
| Gradient (Random Init) | 20.59 ±0.85 | 15.04 ±1.13 | 19.41 ±0.90 | 19.81 ±1.86 | 19.10 ±1.02 | 16.11 ±1.45 | 23.03 ±0.72 | 21.15 ±0.82 |

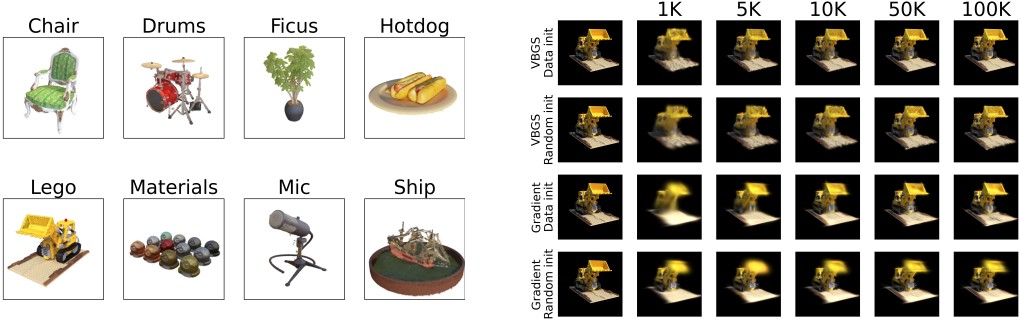

Figure 2: **Mixture model performance on 3D data.** (left) Image reconstructions for each of the 8 objects, given a VBGS with 100k components. (right) Qualitative performance when using various amounts of components for reconstructing "lego" for both VBGS and the gradient approach.

of ellipsoids, while the gradient approach fills the areas more easily. We attribute this to a strong prior over the covariance shape, encoded in the Wishart hyperparameters.

Finally, we also conduct the continual learning experiment for 3D and observe that the same properties from the 2D experiment hold, reaching an average reconstruction error over all objects of $11.19 \pm 3.53$ dB for VBGS (Random Init) and $21.26 \pm 1.76$ dB for Gradient (Random Init). Crucially, in the continual learning setting the model does not have access to the data for initialization, and has to be initialized randomly. For a quantitative evaluation of performance as a function of the amount of observations, see Appendix F.2.

# 4 Discussion and Conclusion

In this paper, we proposed a method for optimizing Gaussian splats using Variational Bayes. We evaluated this method on both a 2D and a 3D dataset and showed that we achieved on-par performance while gaining the advantage that this approach enables continual learning. One of the main limitations of our approach compared to 3DGS is that we require RGBD data, as opposed to optimizing directly on RGB projections. We did not consider a mechanism to grow the model, whereas 3DGS achieves the best results when it can dynamically grow and shrink the model. Future research could investigate a principled approach to dynamically determine model size, as we can measure the model evidence (Friston et al., 2023). While VBGS only requires a single update step, it's important to note that this step is computationally more expensive than a single backpropagation step. This means that in the case of a large dataset that doesn't fit in memory, we are required to split it into distinct update steps and stochastic gradient descent might be faster overall. Of course, one might not need all this data, which could be an interesting avenue of research, i.e. active data selection. We believe that VBGS could open the door toward active learning, e.g. in robot SLAM. Due to continual learning, the model can continuously integrate novel information without the need for a replay buffer (Sucar

et al., 2021; Matsuki et al., 2024). Combining this with the variational posterior over parameters, an embodied agent would be able to do parameter-based exploration (Schwartenbeck et al., 2013), efficiently exploring and building a model of the environment in real-time.

## 5    Acknowledgments and Disclosure of Funding

The authors would like to thank the members of VERSES for critical discussions and feedback that improved the quality of this work, with special thanks to Jeff Beck, Tommaso Salvatori, and Conor Heins.

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

# A    The generative model

VBGS relies on the conjugate properties of exponential family distributions. In this section, we first write down the functional form of these distributions and then describe the particular generative model for representing space and color.

## A.1    The exponential family

If the likelihood distribution is part of the exponential family, it can be written down as:

$$p(x|\theta) = \phi(x) \exp\left(\theta \cdot T(x) - A(\theta)\right), \tag{5}$$

where $\theta$ are the natural parameters of the distribution, $T(x)$ is the sufficient statistic, $A(\theta)$ is the log partition function, and $\phi(x)$ is the measure function. This likelihood has a conjugate prior of the following form:

$$p(\theta|\eta_0, \nu_0) = \frac{1}{Z(\eta_0, \nu_0)} \exp\left(\eta_0 \cdot \theta - \nu_0 \cdot A(\theta)\right), \tag{6}$$

where $Z(\eta_0, \nu_0)$ is the normalizing term. This distribution is parameterized by its natural parameters $(\eta_0, \nu_0)$. The posterior is then of the same functional form as the prior, whose natural parameters can be calculated as a function of the sufficient statistics of the data and the natural parameters of the prior, i.e. $(\eta_0 + \sum_x T(x), \nu_0 + \sum_x 1)$ (Murphy, 2013).

## A.2    The generative mixture model for space and color

The generative model described in Equation (2) is factorized over $N$ data points and $K$ mixture components. The mixture components over space $p(s|z, \mu_s, \Sigma_s)$ are conditionally independent from the component over color $p(c|z, \mu_c, \Sigma_c)$, given mixture component $z$. These random variables are parameterized by the following distributions:

$$z_n \sim \text{Cat}(\pi) \tag{7} \qquad c_n|z_n = k \sim \text{MVN}(\mu_{k,c}, \Sigma_{k,c}) \tag{9}$$
$$s_n|z_n = k \sim \text{MVN}(\mu_{k,s}, \Sigma_{k,s}) \tag{8}$$

The parameters of these distributions are also modeled as random variables. As the conjugate prior of a multivariate normal (MVN) is a Normal Inverse Wishart (NIW) distribution, and of a categorical (Cat) distribution is a Dirichlet distribution, we use the following:

$$\mu_{k,s}, \Sigma_{k,s} \sim \text{NIW}(m_{0,s}, \kappa_{0,s}, V_{0,s}, n_{0,s}) \tag{10}$$
$$\mu_{k,c}, \Sigma_{k,c} \sim \text{NIW}(m_{0,c}, \kappa_{0,c}, V_{0,c}, n_{0,c}) \tag{11}$$
$$\pi \sim \text{Dirichlet}(\alpha_0), \tag{12}$$

We choose the approximate posteriors in the same family as the corresponding priors. Except, for the color $c$, we model the mean by a Normal distribution, and keep the covariance fixed:

$$q(z_n) = \text{Cat}(\gamma_n) \tag{13} \qquad q(\Sigma_{k,c}) = \text{Delta}(\varepsilon I), \tag{16}$$
$$q(\mu_{k,s}, \Sigma_{k,s}) = \text{NIW}(m_{t,s}, \kappa_{t,s}, V_{t,s}, n_{t,s}) \tag{14} \qquad q(\mu_{k,c}) = \text{Normal}(m_{t,c}, \kappa_{t,c}^{-1}\varepsilon I) \tag{17}$$
$$q(\pi) = \text{Dirichlet}(\alpha_t) \tag{15}$$

where the subscript t indicates the parameters at timestep $t$, and $\varepsilon$ is a chosen hyperparameter.

# B    Coordinate ascent variational inference

The approximate posterior is computed by maximizing the evidence lower bound is maximized with respect to the variational distribution $q(z, \mu_s, \Sigma_s, \mu_c, \Sigma_c, \pi)$. Recall the ELBO (Equation (4)):

$$\text{ELBO} = \mathbb{D}_{\text{KL}}[q(z, \mu_s, \Sigma_s, \mu_c, \Sigma_c, \pi) \,||\, p(z, \mu_s, \Sigma_s, \mu_c, \Sigma_c, \pi|s, c)], \tag{18}$$

This is done using coordinate ascent variational inference (Blei et al., 2017), which contains two distinct steps, that are iteratively executed to optimize the model parameters. It draws parallels to the

well-known expectation maximization (EM) algorithm: the first step computes the expectation over assignments $q(z)$ for each of the data points. In the second step, instead of computing the maximum likelihood estimate as is done in EM, we maximize the variational parameters of the posterior over model parameters.

In the first step, the assignment for each data point $(s_n, c_n)$ is computed, by deriving the ELBO with respect to $q(z_n)$.

$$\log q(z_n = k) = \log \gamma_n = \mathbb{E}_{q(\mu_{k,s}, \Sigma_{k,s})}[\log p(s_n | \mu_{k,s}, \Sigma_{k,s})] \tag{19}$$
$$+ \mathbb{E}_{q(\mu_{k,c}, \Sigma_{k,c})}[\log p(c_n | \mu_{k,c}, \Sigma_{k,c})] \tag{20}$$
$$+ \mathbb{E}_{q(\pi)}[\log p(\pi)] - \log Z_n, \tag{21}$$

where $Z_n$ is a normalizing term, i.e. if $\hat{\gamma}_n$ is the unnormalized logit, we can find the parameters of the categorical distribution as $\gamma_n = \frac{\hat{\gamma}_n}{\sum \hat{\gamma}_n}$.

In the second step, we compute the update of the approximate posteriors over model parameters. To this end, we derive the ELBO with respect to the natural parameters of the distributions. If the distribution is conjugate to the likelihood, then we acquire updates of the following parametric form:

$$\eta_k = \eta_{0,k} + \sum_{x_n \in \mathcal{D}} \gamma_{k,n} T(x_n) \tag{22} \qquad \nu_k = \nu_{0,k} + \sum_{x_n \in \mathcal{D}} \gamma_{k,n}, \tag{23}$$

where $\eta$ and $\nu$ are the natural parameters of the distribution over the likelihood's natural parameters, and $T(x_n)$ are the sufficient statistics of the data $x_n$.

For the approximate posterior $q(\mu_{s,k}, \Sigma_{s,k})$ over the parameters of the spatial likelihood, the sufficient statistics are given by $T(s_n) = (s_n, s_n \cdot s_n^T)$. The NIW conjugate prior consists of a Normal distribution over the mean and an inverse Wishart distribution over the covariance matrix. Hence, for each of the prior's natural parameters, it has two values: $\eta_{0,s} = (\kappa_{0,s} \cdot m_{0,s}, V_{0,s} + \kappa_{0,s} \cdot m_{0,s} \cdot m_{0,s}^T)$ and $\nu_{0,s} = (\kappa_{0,s}, n_{0,s} + D_s + 1)$. Here, $m_{0,s}$ is the mean of the Normal distribution over the mean, $\kappa_0$ is the concentration parameter over the mean, $n_{0,s}$ indicates the degrees of freedom, $V_{0,s}$ the inverse scale matrix of the Wishart distribution, and $D_s$ the dimensionality of the MVN.

For the approximate posterior $q(\mu_{c,k}, \Sigma_{c,k})$ over the color likelihood, the sufficient statistics are again given by: $T(c_n) = (c_n, c_n \cdot c_n^T)$. The prior is parameterized similarly as the NIW over $s$: $\eta_{0,c} = (\kappa_{0,c} \cdot m_{0,c}, V_{0,c} + \kappa_{0,c} \cdot m_{0,c} \cdot m_{0,c}^T)$ and $\nu_{0,c} = (\kappa_{0,c}, n_{0,c} + D_c + 1)$. However, as we model the prior over $\Sigma_{k,c}$ as a delta distribution, we keep the values for $n_{k,c}$ and $V_{k,c}$ fixed.

Finally, for the approximate posterior over the $q(\pi)$ over the component assignment likelihood $z$, the conjugate prior is a Dirichlet distribution. Here, sufficient statistics are given by $T(x) = 1$, and the prior is parameterized by the natural parameter $\eta_{0,z} = \alpha$.

## B.1 Continual updates

These update rules lend themselves to continual learning. In this setting, data is processed sequentially. At each time point $t$, a batch of $\mathcal{D}_t$ of data points $(s_n, c_n)$ is available, and the variational posterior over model parameters is updated.

For each of these data points, the assignments $\gamma_{k,n}$ can be calculated similarly to Equation (19). We can then rewrite the update steps from Equations (22) and (23), in a streaming way for $T$ timesteps:

$$\eta_k = \eta_{0,k} + \sum_{t=1}^{T} \sum_{x_n \in \mathcal{D}_t} \gamma_{k,n} T(x_n), \tag{24} \qquad \nu_k = \nu_{0,k} + \sum_{t=1}^{T} \sum_{x_n \in \mathcal{D}_t} \gamma_{k,n} \tag{25}$$

We can then rewrite this as the following iterative updates for the natural parameters, starting at $t = 1$, with the prior value at $t = 0$:

$$\eta_{t,k} = \eta_{t-1,k} + \sum_{x_n \in \mathcal{D}_t} \gamma_{k,n} T(x_n), \tag{26} \qquad \nu_{t,k} = \nu_{t-1,k} + \sum_{x_n \in \mathcal{D}_t} \gamma_{k,n} \tag{27}$$

Where $x_n$ is a placeholder for the particular data, e.g. when updating $q(\mu_{s,k}, \Sigma_{s,k})$, this would be $s_n$.

Note that when $\gamma_{k,n}$ is calculated using the initial parameterization of the variational posterior over parameters, applying these continual updates is identical to processing all the data in a single batch. Hence, the problem of catastrophic forgetting is avoided.

## C  Hyperparameters

### C.1  Prior parameters

The considered conjugate priors over the likelihood parameters are parameterized by the canonical parameters shown in Table 2. Some of these values are a function of the number of available components, indicated by nc.

Table 2: **Parameters of the conjugate priors over likelihood parameters.** Some values are a function of the number of components, indicated by nc, $I$ represents the identity matrix of size the multivariate dimension $D$. Parameters are in the canonical form of the corresponding distribution.

|  |  | **2D** | **3D** |
|---|---|---|---|
| $p(\mu_{k,s}, \Sigma_{k,s})$ | $m_{s,k}$ | $\mathbf{0}$ | $\mathbf{0}$ |
|  | $\kappa_{s,k}$ | $10^{-2} \cdot \mathbf{1}$ | $10^{-2} \cdot \mathbf{1}$ |
|  | $V_{s,k}$ | $2.25 \cdot 10^4 \cdot \text{nc} \cdot I$ | $2.25 \cdot 10^6 \cdot \text{nc} \cdot I$ |
|  | $n_{s,k}$ | $4$ | $5$ |
| $p(\mu_{k,c}, \Sigma_{k,c})$ | $m_{c,k}$ | $\mathbf{0}$ | $\mathbf{0}$ |
|  | $\kappa_{c,k}$ | $10^{-2} \cdot \mathbf{1}$ | $10^{-2} \cdot \mathbf{1}$ |
|  | $V_{c,k}$ | $10^6 \cdot I$ | $10^8 \cdot I$ |
|  | $n_{c,k}$ | $5$ | $5$ |
| $p(\pi)$ | $\alpha_k$ | $\frac{1}{\text{nc}}$ | $\frac{1}{\text{nc}}$ |

### C.2  Initial parameters

In the first step of the coordinate ascent algorithm, $q(z)$ is inferred with an initial configuration of $q(z, \mu_s, \Sigma_s, \mu_c, \Sigma_c, \pi)$. Crucially, this configuration is distinct from the prior described in Appendix C.1. The initial canonical parameters are shown in Table 3.

The values of $m_{s,k\text{init}}$ and $m_{c,k\text{init}}$ vary for the two considered cases. When the means are initialized on the data point (Data Init), $m_{s,k\text{init}}$ and $m_{c,k\text{init}}$ are set to $K$ values $(s_n, c_n)$ sampled from the data $\mathcal{D}$. When the means are randomly initialized (Random Init), then $m_{k,s,\text{init}} \sim \mathcal{U}[-1, 1]$, $m_{k,c,\text{init}} \sim \delta(0)$.

Table 3: **Parameters of the initial approximate posteriors over likelihood parameters.** Some values are a function of the number of components, indicated by nc, $I$ represents the identity matrix of size the multivariate dimension $D$. Parameters are in the canonical form of the corresponding distribution.

|  |  | **2D** | **3D** |
|---|---|---|---|
| $q(\mu_{k,s}, \Sigma_{k,s})$ | $m_{s,k}$ | $\mathbf{m}_{s,k,\text{init}}$ | $\mathbf{m}_{s,k,\text{init}}$ |
|  | $\kappa_{s,k}$ | $10^{-5} \cdot \mathbf{1}$ | $10^{-6} \cdot \mathbf{1}$ |
|  | $V_{s,k}$ | $2.25 \cdot 10^4 \cdot \text{nc} \cdot I$ | $2.25 \cdot 10^6 \cdot \text{nc} \cdot I$ |
|  | $n_{s,k}$ | $4$ | $5$ |
| $q(\mu_{k,c}, \Sigma_{k,c})$ | $m_{s,k}$ | $\mathbf{m}_{c,k,\text{init}}$ | $\mathbf{m}_{c,k,\text{init}}$ |
|  | $\kappa_{s,k}$ | $10^{-2} \cdot \mathbf{1}$ | $10^{-2} \cdot \mathbf{1}$ |
|  | $V_{c,k}$ | $10^6 \cdot I$ | $10^8 \cdot I$ |
|  | $n_{c,k}$ | $5$ | $5$ |
| $q(\pi)$ | $\alpha_k$ | $\frac{1}{\text{nc}}$ | $\frac{1}{\text{nc}}$ |

## D Image rendering

Rendering is the process of generating an image, given an internal representation. Typically, this refers to the process of projecting from a 3D representation to the image plane. In our generative model, this process boils down to evaluating the expected color value for each considered pixel, i.e. $\mathbb{E}_{p(c|s)}[c]$. This is straightforward to compute for image data:

$$\mathbb{E}_{p(c|s)}[c] = \sum_k \left( p(z_n = k|s) \sum_{c_i} c_i \underbrace{p(c_i|z_n = k)}_{\approx \delta(c_k)} \right) \approx \sum_k p(z_n = k|s)c_k, \tag{28}$$

where we approximate the distribution over the color features by a delta distribution positioned at the mean of $q(\mu_c)$.

In equation (28), the distribution is conditioned on the spatial location of the pixel. For rendering in 3D, we leverage the computationally efficient 3D renderer designed by Kerbl et al. (2023). Here, the 3D Gaussians are first projected to the image plane, and by using alpha blending along a casted ray color values are combined into a pixel color. As we do not optimize on image reconstruction, our approach does not have sensible alpha blending. We therefore set the alpha value for all Gaussians at 1, i.e. all components are opaque.

## E Data Normalization

Before training, the data is normalized to have zero mean and a standard deviation of one. This is not strictly necessary but ensures that we can use the same hyperparameters and initial parameters for all models. When all data is readily available, the data is simply normalized using the statistics from the data itself. Note that each dimension is considered individually.

When the data is not available, we assume that the random variable is distributed uniformly within a certain range: $x \sim \mathcal{U}(r_{\min}, r_{\max})$. The normalized value is then calculated as:

$$\hat{x} = \frac{x - \mathbb{E}[x]}{\sqrt{\mathrm{Var}[x]}}, \tag{29}$$

for which the ranges for each of the parameters are displayed in Table 4.

Table 4: **Range for data normalization.** The subscript i indicates a single dimension of the vector.

|       | Range 2D | Range 3D |
|-------|----------|----------|
| $s_i$ | $[0, 64]$  | $[-1, 1]$  |
| $c_i$ | $[0, 255]$ | $[0, 255]$ |

## F Additional results

### F.1 Reconstruction performance as a function of number of components

Figure 3a shows the reconstruction of an image from the TinyImageNet dataset using various model sizes for both VBGS and the gradient model. Notice how in low component regimes, the initiation on data yields smoother gaussian components (e.g. in the 100 components regime), however when the capacity goes up, this impact is drastically reduced. In high component regimes, the gradient approach yields a smoothed version of the surface, while the high frequency textures are better captured by the VBGS models.

We evaluated reconstruction performance on 3D for a variety of models, as a function of model size, measured as the amount of available components. The results are shown in Figure 3b.

### F.2 Continual learning results

In this section, we evaluate the performance of VBGS in the continual learning setting for both the 2D and 3D datasets. For the image dataset, we divide the image in patches of 8 by 8 pitches, and feed

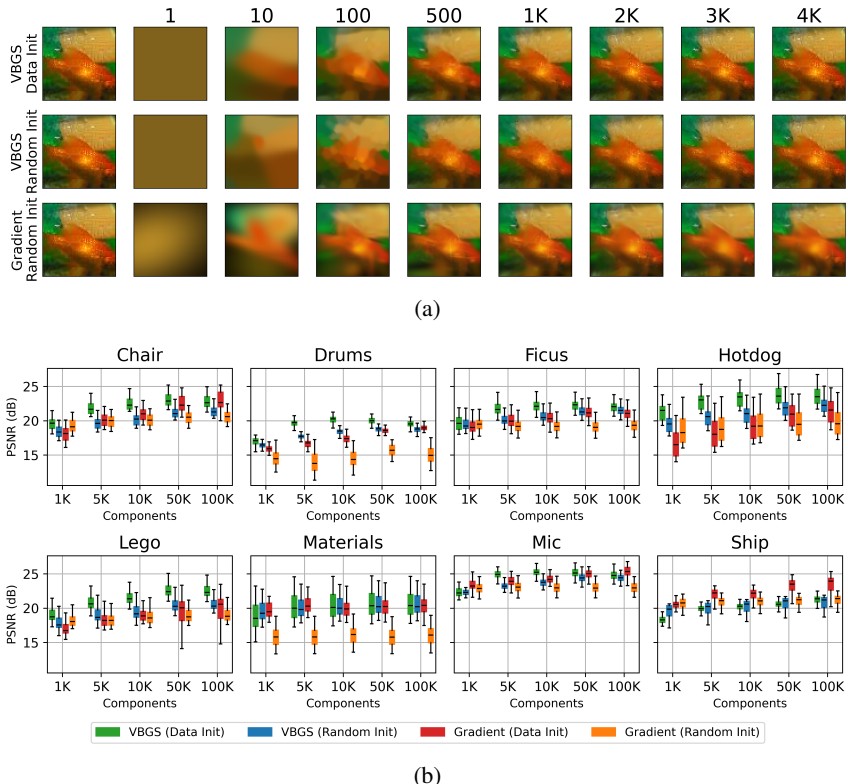

(a)

(b)

Figure 3: **Additional results for reconstruction performance**. (a) Qualitative results for image reconstruction as a function of model size. (b) Reconstruction performance for 3D models as a function of model size. Evaluated on the validation set for the 8 objects from the Blender dataset (Mildenhall et al., 2020).

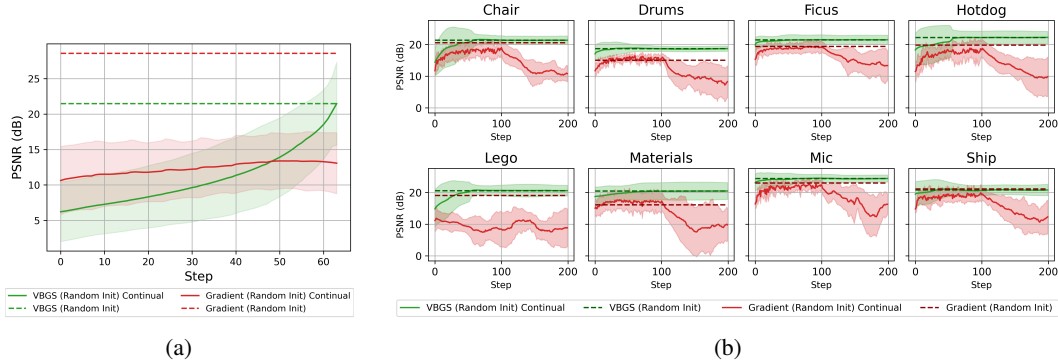

(a)

(b)

Figure 4: **Additional results for continual learning**. (a) Evolution of reconstruction performance measured as PSNR (dB) after feeding image patches of size 8x8 sequentially to the model. Red indicates the VBEM approach, while blue indicates the gradient approach. The performance achieved by feeding in all data at once is denoted by the dashed line. Confidence intervals are the $Z_{95}$ interval computed over the 10k validation images of the Tiny ImageNet dataset of size 64x64. (b) Evolution of the reconstruction performance, measured as PSNR (dB), after feeding in consecutive images of an object. The performance achieved by feeding in all data at once is denoted by the dashed line. Confidence intervals are the $Z_{95}$ interval computed over the 100 frames from the validation set.

in a single patch at each time step as shown in Figure 1 (right). VBGS applies a single update for each received patch, while for the Gradient approach we do 100 training steps with a learning rate 0.1.

Figure 4a shows how the reconstruction PSNR evolves at each time step over the tiny ImageNet test set for the model with capacity 10K components. The shaded area highlights the $Z_{95}$ confidence interval. It's important to note that the reconstruction error of VBGS after observing all the patches converges to the same value as observing the whole image at once, as the inferred posterior ends up being identical. Even though the gradient approach when observing all data at once achieves much higher PSNR, when the data is fed in continuously, it does not achieve that performance in the continual setting, as it always focuses on the latest observed patch (see Figure 1 (right)).

In the case of the 3D dataset, we feed in a continuous stream of images, similar to the image experiment. For training: in VBGS we do a single update for each observed (RGBD) image, and for the gradient baseline we do 100 gradient steps with learning rate 0.1. As we are now dealing in 3D, we evaluate on novel-view prediction. The models are trained on the blender test set of 200 continuous frames with depth information, and evaluated on the validation set of 100 images. Figure 4b shows the evolution of PSNR, again with the $Z_{95}$ interval marked in the shaded area. Note how the performance of the gradient approach deteriorates after observing more and more frames. This figure also indicates how, even though 200 frames are provided, VBGS reaches that level of performance a lot quicker, i.e. after 50 steps.

## F.3 Growing and shrinking of 3D Gaussian Splatting

Table 5: **Dynamic 3DGS.** The first row shows reconstruction performance, measured in PSNR (dB), evaluated on the validation set of the 8 objects of the Blender set. The second row for each model, shows the resulting amount of components after optimization.

|  | chair | drums | ficus | hotdog | lego | materials | mic | ship |
|---|---|---|---|---|---|---|---|---|
| Gradient (Data Init) | 26.73 ±1.15 | 20.65 ±0.71 | 22.93 ±1.09 | 27.92 ±0.84 | 26.94 ±1.10 | 17.02 ±1.48 | 28.10 ±0.81 | 25.91 ±1.62 |
|  | 456K | 378K | 281K | 177K | 337K | 46K | 186K | 263K |
| Gradient (Random Init) | 26.72 ±1.16 | 20.65 ±0.69 | 22.94 ±1.10 | 28.06 ±0.80 | 26.98 ±1.11 | 17.03 ±1.48 | 28.26 ±0.79 | 25.81 ±1.59 |
|  | 455K | 384K | 282K | 177K | 339K | 53K | 182K | 262K |

3D Gaussian Splatting in its default implementation dynamically grows and shrinks the model. In this paper, we kept the number of components fixed, to make a comparison as a function of model size. Here, we optimize 3D Gaussian Splats using the gradient based approach with dynamic model sizes. The results are reported in Table 5. We can see that the achieved reconstruction quality is much higher, but they also require a larger amount of components.

