# OpenReview forum: "Variational Bayes Gaussian Splatting"
_NeurIPS.cc/2024/Workshop/BDU — NeurIPS BDU Workshop 2024 Poster_

### Official Review · Reviewer_bKBs · 2024-09-14
**Assessment of a Closed-Form Update Rule for Variational Posterior: Strengths, Methodological Details, and Suggestions**

**Rating:** 7
**Confidence:** 4

**Review:**

The author found a closed-form update rule for the variational posterior, which allows us to continually apply updates from partial data, using only a single update step for each observation. The overall quality of the article is excellent, no replay buffer is required to reduce the number of data reads and writes. By storing data to a certain extent before transferring it, the number of transfers can be significantly reduced, thus saving time. The deduction process is reasonable, the number of components may affect the accuracy and stability of the model and should be further explained in detail.

Pros:
1. The author found a closed-form update rule for the variational posterior, which allows us to continually apply updates from partial data, using only a single update step for each observation. The overall quality of the article is excellent, no replay buffer is required to reduce the number of data reads and writes.

Cons:
1. By storing data to a certain extent before transferring it, the number of transfers can be significantly reduced, thus saving time. The deduction process is reasonable, the number of components may affect the accuracy and stability of the model and should be further explained in detail.
2. In the conclusion, in addition to summarizing the actions taken and results, please strengthen the explanation of their significance. It is recommended to use quantitavite reasoning comparing with appropriate benchmarks, especially those stemming from previous work.

---

### Official Review · Reviewer_trhS · 2024-09-18
**The paper introduces Variational Bayes Gaussian Splatting (VBGS), a novel approach to 3D scene representation using variational inference. VBGS enables continual learning without replay buffers, outperforming gradient-based methods in efficiency and sequential data handling. Experimental results on 2D and 3D datasets demonstrate its effectiveness in scene reconstruction.**

**Rating:** 7
**Confidence:** 3

**Review:**

The paper is of high quality and presents a clear, well-structured exposition of VBGS. Its originality lies in the application of variational inference to Gaussian splatting for 3D scene reconstruction, which is a novel contribution to the field. The proposed closed-form update rule enables continual learning without catastrophic forgetting, which addresses a significant limitation of traditional gradient-based methods.

The clarity of the paper is commendable, with detailed derivations of the variational update rules and comprehensive comparisons to gradient-based approaches. The experimental results on Tiny ImageNet and the Blender dataset are thorough and demonstrate the method's effectiveness.

One potential limitation is the computational cost of each update step in VBGS, which, although more efficient than gradient-based methods in some cases, could become burdensome with very large datasets. Overall, the paper provides significant contributions to 3D scene representation and continual learning, making it a valuable addition to the field.

---

### Decision · Program_Chairs · 2024-10-09

Accept (Poster)